# Searching for a Numerical Model for Prediction of Pressure-Swirl Atomizer Internal Flow

**Milan Maly** [1,*] , **Jaroslav Slama** [2] , **Ondrej Cejpek** [1] **and Jan Jedelsky** [1]

1. Energy Institute, Brno University of Technology, Technicka 2, 616 69 Brno, Czech Republic; ondrej.cejpek@vutbr.cz (O.C.); jedelsky@fme.vutbr.cz (J.J.)
2. Provyko s.r.o, Vinarska 460, 603 00 Brno, Czech Republic; jaroslav.slama@provyko.cz
* Correspondence: author: milan.maly@vutbr.cz

**Abstract:** Numerical prediction of discharge parameters allows design of a pressure-swirl atomizer in a fast and cheap manner, yet it must provide reliable results for a wide range of geometries and operating regimes. Many authors have used different numerical setups for similar cases and often concluded opposite suggestions on numerical setup. This paper compares 2D (two-dimensional) axisymmetric, 3D (three-dimensional) periodic and full 3D numerical models used for estimation of the internal flow characteristics of a pressure-swirl atomizer. The computed results are compared with experimental data in terms of spray cone angle, discharge coefficient ($C_D$), internal air-core dimensions, and velocity profiles. The three-component velocity was experimentally measured using a Laser Doppler Anemometry in a scaled transparent model of the atomizer. The internal air-core was visualized by a high-speed camera with backlit illumination. Tested conditions covered a wide range of the Reynolds numbers within the inlet ports, $Re$ = 1000, 2000, 4000. The flow was treated as both steady and transient flow. The numerical solver used laminar and several turbulence models, represented by $k$-$\varepsilon$ and $k$-$\omega$ models, Reynolds Stress model (RSM) and Large Eddy Simulation (LES). The laminar solver was capable of closely predicting the $C_D$, air-core dimensions and velocity profiles compared with the experimental results in both 2D and 3D simulations. The LES performed similarly to the laminar solver for low $Re$ and was slightly superior for $Re$ = 4000. The two-equation models were sensitive to proper solving of the near wall flow and were not accurate for low $Re$. Surprisingly, the RSM produced the worst results.

**Keywords:** pressure-swirl; internal flow; laminar; CFD; LDA; LES; two-phase; velocity

## 1. Introduction

Pressure-swirl atomizers (PS) play a unique role in many industrial applications including combustion, spray cooling, spray drying, etc. A relatively simple geometrical design, resistance to clogging, wide spray cone angle and high atomization efficiency are among the favourable parameters. A typical PS atomizer contains tangential entry ports and a swirl chamber with an exit orifice. The liquid is fed via tangential ports into the swirl chamber where it gains high swirl velocity. The swirling liquid is discharged from the exit orifice in the form of a hollow conical liquid sheet which consequently disintegrates into filaments and ligaments. A ratio of the swirl to axial momentum determines the spray cone angle (*SCA*). Despite the simple geometry, the internal flow behaviour is complex. The swirling liquid creates a low-pressure zone along a centre line of the swirl chamber where the static pressure usually drops below the ambient pressure, and the air from the surrounding atmosphere is pulled inside the low-pressure zone, so an air-core is formed. The internal vortex behaves as a Rankine vortex since the swirl velocity has its maximum located at the air-core surface which behaves like a virtual solid cylinder [1]. Secondary flow effects, as Görtler vortices, could be also presented in a boundary layer inside the swirl chamber [2,3].

It is well known that the internal flow directly affects the parameters of the discharged liquid sheet, such as its thickness, stability, velocity, and *SCA*. These parameters consequently determine the sizes of ligaments and droplets. To understand the link between the atomizer's performance and its design, the internal flow must be examined. Some authors used the exact analytical solution to predict the discharge parameters. Simple non-viscous treatment, reviewed in [4,5], proved to be a useful tool for a basic insight into the flow behaviour, but it lacks accuracy for some atomizer geometries. A better agreement can be achieved when the viscous flow is assumed [6,7], but still some aspects such as the liquid sheet temporal stability or secondary flow effects, are not resolved and their mathematical description is very extensive.

On the other hand, numerical simulation has arisen in recent years due to an increase in computational performance, and many commercial software programs are available in the market. These software tools for numerical simulations of the flow dynamics could be simply applied to predicting the internal flow of the PS atomizer. However, many different geometrical, numerical and physical setup combinations can be used for the same atomizer and operating conditions. The CFD simulation of the internal flow ideally suits for final adjustments of the atomizer geometry, since it allows for fine-tuning of the atomizer's individual parts and captures the internal flow instabilities. Nevertheless, the numerical results should be still validated by an experiment.

In the past, many authors performed a CFD simulation of the internal flow of PS atomizers. An overview of some published papers is presented in Table 1. One of the first numerical studies of the PS atomizer was conducted in 1997 by Yule and Chinn [8]. They used a 2D axisymmetric geometry with a laminar solver and reported a deviation of discharge coefficient, $C_D$, from experimental data to be less than 3%. A similar numerical setup was later used by Amini [6]. Even this simple 2D model yield better agreement with the experimental data than the analytical viscous solution. Note here that the author used the laminar solver even for values of Reynolds number within the inlet port, $Re = 122,000$. The port-based $Re$ is defined as $Re = v_p d_p \rho_l / \mu_l$, where $v_p$ is velocity within the inlet port, $d_p$ is the inlet port hydraulic diameter, $\mu_l$ is liquid dynamic viscosity and $\rho_l$ is liquid density. The complex nature of the internal flow does not allow for a simple conclusion about whether the flow is turbulent or laminar, and many authors claim the opposite. A theoretical evaluation of turbulence evolution within the swirl chamber was made by Yule and Chinn [9,10] who suggest that the flow is laminar even for very high $Re$ due to the laminarization effect of the swirl dominant flow itself and also due to the very short length scale where the turbulence has no time to develop.

A comparison of Large Eddy Simulation (LES) and laminar models was performed by Madsen et al. [11]. They used a scaled atomizer and operated it in the range of $Re = 12,000–41,000$. Under these operating regimes, the laminar model had a slightly better agreement with the experimental data than LES. The authors also examined simple turbulence models represented by RNG (renormalization group) and realizable $k$-$\varepsilon$ models. However, these models were unable to predict the internal air-core. Galbiati et al. [12] compared the LES simulation with RNG $k$-$\varepsilon$ and RSM (Reynolds Stress model). They found an insignificant variation among the used models when compared to the deviation in the results from published empirical correlations. They also noted that the flow field was consistent for the LES and $k$-$\varepsilon$ model, while, surprisingly, the RSM had some discrepancies. Note here that the authors claim that the internal flow is fully turbulent based on Walzel Reynolds number, $Re_w$, defined as: $Re_w = \sqrt{2\rho_l p_l} d_o / \mu_l$, since it was much larger than a critical value of 5000. However, the calculated $Re = 1700–3800$ rather suggests laminar flow. Qian [3] found RSM to be superior over the laminar model at $Re = 16,000$, while Nouri-Borujerdi [13] found opposite conclusions for even larger $Re$ in the range of 18,000–40,000.

Baharanchi et al. [14] examined several schemes to capture the liquid–air interface using a 2D simulation with the RNG $k$-$\varepsilon$ turbulence model. A geo-reconstruct scheme was found to be the most suitable for capturing the air-core. They also discussed the necessity

to include surface tension and found that it has an effect only for a Weber number smaller than 204. From a practical point of view, it had a negligible effect on the developed flow. However, it affects the flow development process.

The difference between 2D and 3D computational models was examined by Sumer et al. [15]. They used the laminar solver and found that the air-core diameter was about 5% smaller in the case of the 3D model. However, the frequency of waves on the air-core surface was predicted by both models closely and in good agreement with the experimental data. Vashahi et al. [16] compared RANS (Reynolds-averaged Navier–Stokes) models $k$-$\varepsilon$ and $k$-$\omega$. The $k$-$\omega$ outperforms the $k$-$\varepsilon$ in the *SCA* prediction and convergence speed.

**Table 1.** Review of published numerical setups.

| Author | Software | 2D/3D | Transient /Steady | Turbulence Model | Interface | *Re* |
|---|---|---|---|---|---|---|
| Shaikh [17] | Fluent | 2D, 3D | | Laminar | VOF Geo-rec. | 1000 |
| Laurila [18] | OpenFOAM | 3D Full | Transient | Implicit LES | VOF Geo-rec. | 420–5300 |
| Galbiati [12] | | 3D Full | Transient | LES, RSM, $k$-$\varepsilon$ RNG | VOF | 1700–3800 |
| Ghate [19] | Fluent | 2D | Transient | RSM | VOF PLIC | $7 \times 10^3$–$2 \times 10^4$ |
| Ibrahim [20] | Fluent | 2D | | RSM | VOF Geo-rec | $5 \times 10^3$–$5 \times 10^4$ |
| Amini [6] | Fluent | 2D | Steady | Laminar | VOF | $10^4$–$10^5$ |
| Dikshit [21] | | 3D | Steady | $k$-$\varepsilon$ | VOF | $\sim 10^4$ |
| Madsen [11] | Fluent, CFX | 3D | | Laminar, LES | VOF | $10^4$–$4 \times 10^4$ |
| Bazarov [22] | CFX | 3D | Steady | $k$-$\varepsilon$ | VOF | $>10^4$ |
| Sumer [15] | Fluent | 2D, 3D | Transient | Laminar | VOF HRIC | $1.2 \times 10^4$ |
| Qian [3] | OpenFOAM | 2D | | Laminar, RSM | VOF LS coupled | $\sim 1.6 \times 10^4$ |
| Nouri-Borujerdi [13] | | 2D | Steady /Transient | Laminar, RSM | Level set | $1.8 \times 10^4$–$4 \times 10^4$ |
| Marudhappan [23] | Fluent | 2D, 3D | | Laminar | VOF HRIC | $\sim 2 \times 10^4$ |
| Baharanchi [14] | | 2D | Transient | $k$-$\varepsilon$ RNG | VOF | $\sim 2.5 \times 10^4$ |
| Mandal [24] | Fluent | 2D | | Laminar | VOF | $\sim 4 \times 10^4$ |
| Vashahi [16] | CCM+ | 3D | Steady | $k$-$\varepsilon$, $k$-$\omega$ | VOF HRIC | |

From the overview in Table 1, no clear conclusion can be made on the physical model selection, as both laminar and turbulence approaches were used by different authors for the entire range of *Re*. The most commonly used turbulence models are from the $k$-$\varepsilon$ family and represented by realizable and renormalization group theory, RNG, models. The RSM is also widely used; the authors reported contradictory conclusions. The air-core interface is usually captured by Volume of Fluid (VOF) models.

Both the 2D and 3D simulations seem to provide accurate results; however, no comparison of the periodic 3D and full-scale 3D mesh was found. The majority of the 3D simulations used meshes, which contain tangential ports. This slows the mesh creation process since the cells near the wall are usually very skewed. To overcome this, a no-port version is introduced here with the aim to provide a simpler and faster mesh creation process. Moreover, steady and transient approaches are compared here. This paper aims to provide an overview of the numerical setup for CFD prototyping of small pressure-swirl atomizers. The data presented here are validated by a comprehensive experiment performed in a wide range of *Re* from 1000 to 4000, which corresponds to a 16-times increase

in the inlet pressure, $p_l$. The experimental data in raw form are available as supplementary material to this paper and can be used for the validation of numerical or analytical models.

## 2. Experimental and Numerical Setup

The experiments were performed at a specially designed facility for cold atomizer testing at the Brno University of Technology, Czech Republic. A similar experimental setup was also used in our previous study where it is described in greater detail [25].

### 2.1. The Atomizer Design and Test Bench

The atomizer geometry was derived from a small-sized atomizer studied in our previous work [26], see schematic drawing in Figure 1. Three inlet ports are used to take advantage of the circumferential periodicity of the atomizer. The swirl chamber has a conical converging part to aid manufacturing simplicity. The atomizer is manufactured as ten times scaled copy since the small dimension of the original atomizer does not allow for direct optical measurement. The transparent parts of the atomizer are made from cast polymethyl methacrylate, PMMA, which was ground and polished to achieve transparency. The operating conditions were partially derived from [26] where the original atomizer operated with JET A-1 at $p_l$ = 0.5 MPa roughly yielding $Re$ = 1000, which is at the lower end of the typical operating range. Other operating regimes, see Table 2, covered $Re$ = 2000 and 4000, which is close to maximal pressure with respect to atomizer body strength. This range of $Re$ values corresponds to the wide range of $p_l$ from 0.5 to 8 MPa for the originally sized (10-times smaller) atomizer using JET A-1.

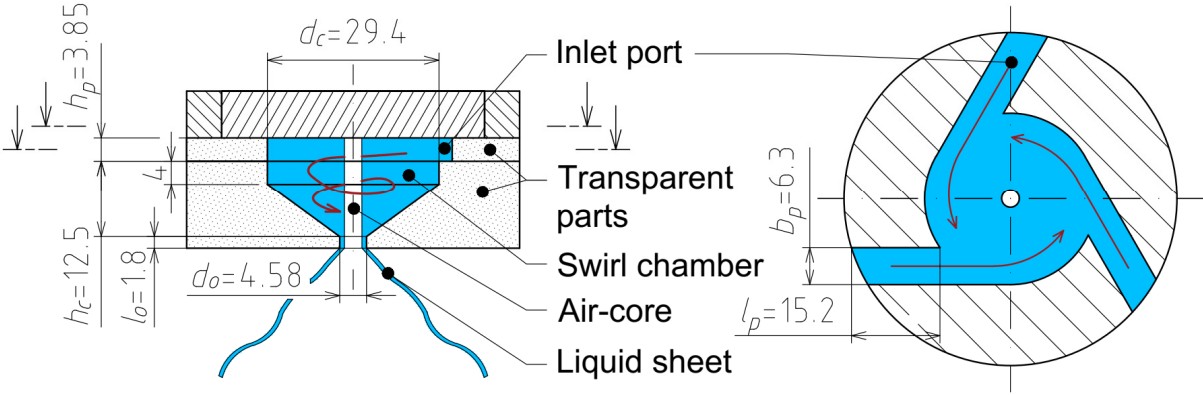

**Figure 1.** Atomizer schematic drawing with main dimensions in millimetres.

**Table 2.** List of operating regimes.

| $Re$ | $p_l$ | $\dot{m}_p$ | $v_p$ | $C_D$ | $Fr$ | $d_{aco}$ | $d_{acc8}$ |
|---|---|---|---|---|---|---|---|
| [–] | [kPa] | [kg/h] | [m/s] | [–] | [–] | [mm] | [mm] |
| 1000 $\pm$ 10 | 2.1 | 47.5 | 0.206 | 0.420 | 5.0 | 2.82 | 1.55 |
| 2000 $\pm$ 10 | 8.8 | 94.8 | 0.411 | 0.410 | 9.9 | 2.99 | 1.64 |
| 4000 $\pm$ 10 | 36.3 | 188.4 | 0.816 | 0.401 | 19.6 | 3.20 | 1.74 |

The operating liquid was p-cymene (1-Methyl-4-(propan-2-yl) benzene). It is one compound, colourless liquid, whose value of refractive index $n$ = 1.49 is closely matched to the atomizer body which simplifies the optical measurement and reduces the measurement errors. The physical properties of the p-cymene at room temperature are as follows: surface tension $\sigma$ = 0.028 kg/s$^2$, liquid dynamic viscosity $\mu_l$ = 0.00085 kg/(m·s), and liquid density $\rho_l$ = 848 kg/m$^3$. The identical test bench as in [25] was used. The uncertainty in pressure sensing was 0.05 kPa and the calculated uncertainty of $C_D$ was 0.25% of the actual value.

## 2.2. High-Speed Imaging

A FASTCAM SA-Z high-speed camera (Photron, Japan) with long-distance microscope 12X Zoom lens (NAVITAR, New York, NY, USA) which is composed of a 2X F-mount adapter (type 1-62922), a 12 mm F.F zoom lens (type 1-50486), and 0.25X lens (type 1-50011) was used to document the spatial and temporal behaviour of the air-core and discharged liquid sheet in one image with spatial dimensions of 31 × 31 mm. The camera frame rate was 20,000 frames per second, the resolution was 1024 × 1024 px and the shutter speed was set to 40 μs. A backlight illumination of the atomizer was provided using an LED panel. The air-core dimensions and the spray cone angle, SCA, were measured by an in-house MATLAB code using a threshold-based detection technique.

## 2.3. Laser Doppler Anemometry

Point-wise velocity measurements inside the swirl chamber were carried out using a 2D LDA FlowExplorer (Dantec Dynamics A/S, Skovlunde, Denmark), for setup details see [25]. The measurements were performed in four axial distances from the atomizer cap, see Figure 2. The distance of $d_{ac2}$, $d_{ac8}$, $d_{ac11}$ and $d_{ac13}$ from the top wall of the swirl chamber is 2, 8, 11 and 13 mm, respectively. The refractive index of the atomizer body and working liquid differed by less than 0.005 for both wavelengths involved in the measurement. Therefore, the velocity error is expected to be less than 0.5% [14,22]. Nevertheless, the real position of the measurement volume inside the atomizer body had to be corrected as $S_2 = nS_1$, where $S_2$ is the real distance of the measurement volume from the atomizer wall and $S_1$ is the traversed distance of the measurement volume from the atomizer wall. The measurements near the air-core surface were affected by light reflections, which generate noise on the LDA signals and subsequent velocity estimates. Therefore, these data were processed by a filtration algorithm which seeks a Gaussian distribution in the velocity histogram and calculates the mean velocity only from the data which satisfies the Gaussian distribution.

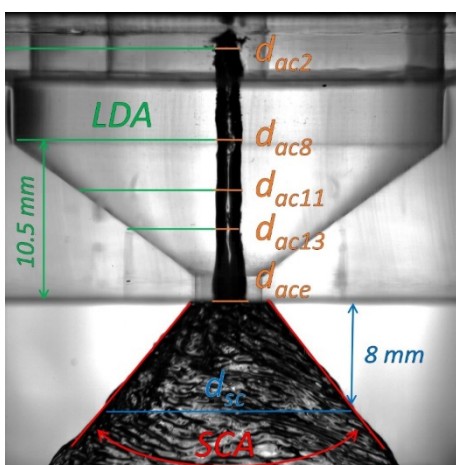

**Figure 2.** The typical high-speed image with measured positions.

Particles SL75 e-spheres with a mean diameter of 45 μm were used as flow tracers for the LDA measurements. Their Stokes number, based on the highest swirl velocity and air-core diameter, which is the worst scenario case for the particle movement, was less than 0.3 for the *Re* = 4000 regime, which ensured a sufficient flow fidelity. Measurement repeatability error was lower than 2% for swirl velocity.

## 2.4. Numerical Setup

The CFD simulations were made using commercial software Ansys Fluent 19.2. Four basic geometries, each representing a different level of simplification, were used: a 2D axisymmetric model, a 3D periodic model without inlet port, a 3D periodic model with

inlet port and a full-scale 3D model with three inlet ports. The 2D axisymmetric model with the swirl component is the simplest approach but is widely used, see Figure 3 left. Note here that the inlet velocity has to be set to conserve the mass flow rate in the radial direction and conserve the angular momentum in the tangential direction. As it is later shown in this paper, this approach underestimates the swirl velocity magnitude. Thus, the second option used here is to set the mean velocity inside the inlet port, $v_p$, as the swirl velocity.

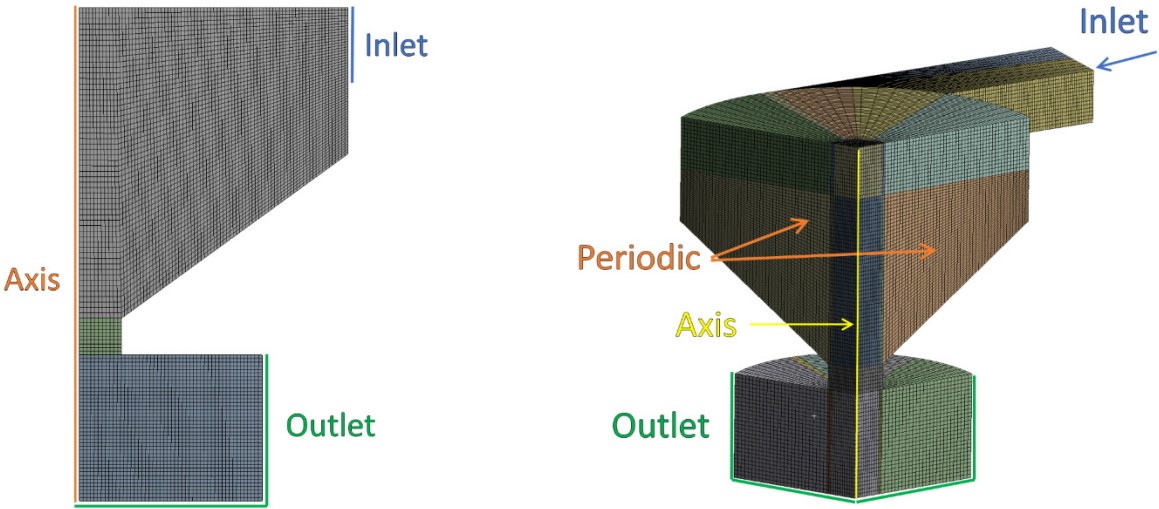

**Figure 3.** (**Left**): The 2D mesh. (**Right**): The 3D periodic mesh with inlet port.

The geometrical design of the atomizer allows for the use of periodic boundary conditions, since the atomizer can be divided into three identical parts, each 120° section. In this way, the 3D periodic model was created. Two periodic models are compared. The first one contains the inlet port (Figure 3 right) and the second one is without the inlet port, a no-port version. The main advantage of the no-port version is simplicity in the mesh generating process. The tangential connection of the port creates skewed elements near the swirl chamber wall. This could be overdone by slight displacement of the port towards the atomizer centreline or using the unstructured tetrahedral or polyhedral mesh. The pressure outlet was set to the outer boundaries and the no-slip condition was used on the internal wall boundaries for all cases. For the turbulent models, the value of turbulence intensity of 1% and hydraulic diameter of 4.9 mm were set on the inlet.

Pressure–velocity coupling was performed using the PISO scheme for the transient solution and the pseudo transient Coupled scheme for the steady solution. Turbulence and momentum used Second-Order Upwind discretization. The liquid–air interaction was captured by a Volume of Fluid model with a geo-reconstruct scheme for transient models or a Compressive scheme for the steady cases and LES. Surface tension between air and liquid was set as a constant value. The air was treated with constant density. The gravity force was also considered.

The transient solution used variable time stepping with a Courant number of 0.15. A typical time step size was approximately $2 \times 10^{-6}$ s. After reaching a quasi-static solution, time averaging was applied, with a minimum of 0.1 s recorded.

Several turbulence models based on a Reynolds-averaged Navier–Stokes (RANS) equation, LES and laminar solver, which solves directly the Navier–Stokes equations, were used and compared to achieve results comparable to the experiment. The governing equations of the used methods and models are well known and can be found elsewhere [27].

Simple two-equation models represented by $k$-$\varepsilon$ and $k$-$\omega$ were chosen for their good accuracy and versatile use for industrial applications. These models determine a turbulent length scale and a time scale by solving two separate transport equations. The $k$-$\varepsilon$ model is based on a transport equation for turbulent kinetic energy $k$ and dissipation rate $\varepsilon$. In this paper, the RNG and realizable $k$-$\varepsilon$ models were used. The wall treatment was performed

using the scalable wall function for meshes without a boundary layer and Enhanced wall treatment (WT) was used for meshes with a boundary layer. The $k$-$\omega$ SST model with low $Re$ correction was used with meshes with boundary layer only. This model combines the standard $k$-$\omega$ model for near wall treatment and the standard $k$-$\varepsilon$ model in the free stream flow.

The Reynolds Stress model (RSM) is among the most advanced RANS models for the swirl dominant flows as it solves all the transport equations for the Reynolds stresses. This model was used with low $Re$ and shear flow correction, with scalable wall function for meshes without prismatic layer and omega-stress based for mesh with the boundary layer.

Large Eddy Simulation (LES) was represented by the Wall-Adapting Local Eddy Viscosity model (WALE). No perturbations were set at the velocity inlet.

## 3. Results and Discussion

In the results, the experimental data are presented first. Then, various numerical approaches are compared. The mesh dependence analysis is followed by a possibility of geometry simplification. Then, several physical models are compared with the experimental data for various $Re$. The final part deals with the air-core temporal behaviour.

### 3.1. Experimental Data

The high-speed imaging showed that the internal air-core is fully developed in all tested regimes, cylindrically shaped, and being larger in diameter at the exit orifice which is a well-known phenomenon [6,28], see Figure 4. With increasing $Re$, the air-core diameter slightly increases as discussed in [7], for diameter values see Table 2, where both $d_{ace}$ and $d_{ac11}$ grow almost linearly with increasing $Re$. The prominence of the surface waves and distortions decreases with $Re$ but the distortions of the discharged liquid sheet increase.

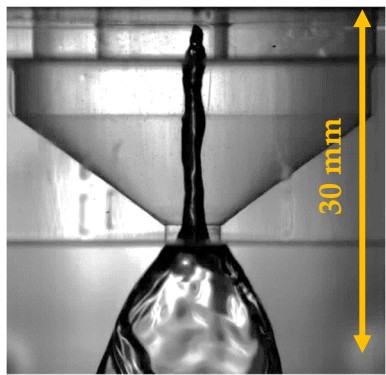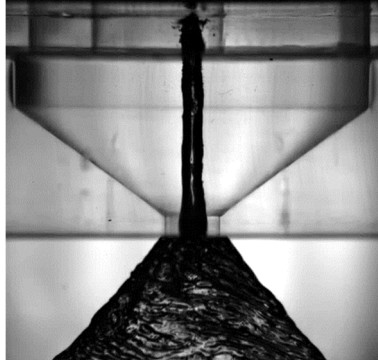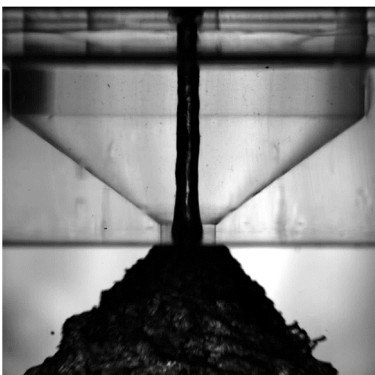

**Figure 4.** Typical results obtained from HS imaging. From left: $Re$ = 1000, 2000 and 4000.

The measured swirl velocity profiles are identical in all axial positions, see Figure 5. This is expected since it is a basic assumption of all inviscid models. The velocity profile represents a Rankine vortex; however, the viscous losses reduce its peak velocity. The measured velocity can be interpolated by a simple equation $wr^b = const$, where $b$ is an empirical constant which is proportional to the viscous losses [29].

With increasing $Re$, the relative swirl velocity slightly increases, which suggests smaller viscous losses with higher $Re$, see increasing values of $b$ constant from 0.8 to 0.9 for $Re$ = 1000 and 4000, respectively. Increasing relative swirl velocity causes an increase in the air-core diameter. This is in alignment with decreasing $C_D$, as shown in Table 2, which also well correlates with the enlarging air-core.

The axial velocity was measurable only in positions $d_{ac8}$ and $d_{ac11}$ due to limited optical access. It has a local maximum near the atomizer wall and a global maximum at the air-core surface (Figure 6). The position and relative value of both maxima remain the same for all measured regimes.

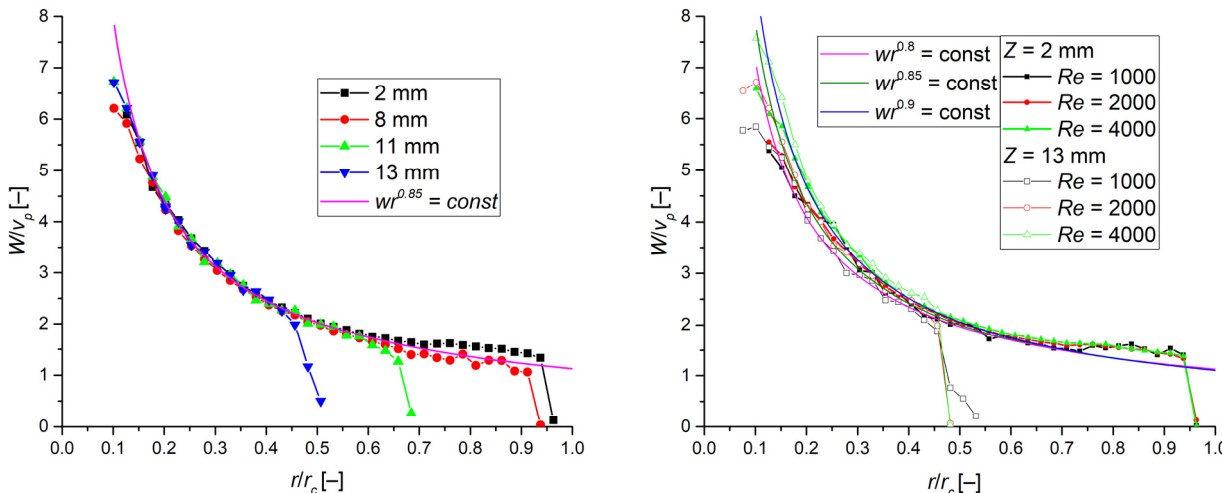

**Figure 5.** Swirl velocity profiles from LDA measurement, left for *Re* = 2000 and various *Z* positions. Right for *Z* = 2 and 13 mm for various *Re*.

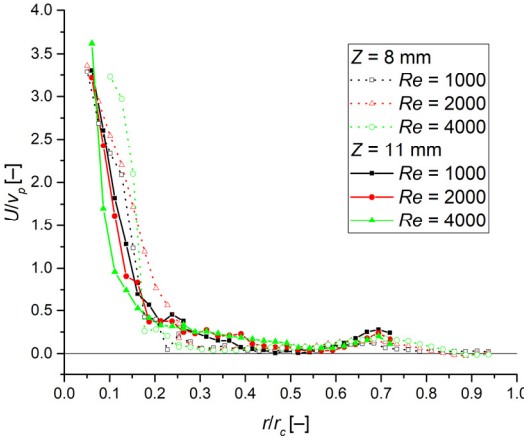

**Figure 6.** Axial velocity profiles from LDA measurements.

The turbulence intensity of the swirl velocity is between 7 and 9% with a slightly decreasing trend with *Re*, but with much higher values near the air-core and swirl chamber walls where it reaches the values above 30%. It was found independent of the axial position. The small influence of *Re* suggests that no transition effect occurs in the studied range of *Re* values between 1000 and 4000. However, the values of the turbulence intensity must be taken with respect to the long measuring volume of the backscatter LDA, which may introduce a false indication of turbulence flow character.

### 3.2. Mesh Independence Test and Comparison of Steady and Transient Models

Several different mesh sizes were compared at *Re* = 2000 using the laminar solver for both steady and transient simulation and steady RANS simulation using a realizable *k-ε* model with scalable wall function. The key parameter that determines the simulation accuracy is the number of mesh cells across the exit orifice; see typical results in Figure 7. With a low number of cells, the error of estimating the air-core dimension rises, which may harm the overall result of the simulation. The effect of cell number on $C_D$, *SCA* and the air-core dimension is shown in Figure 8 for 2D transient and 3D steady and transient simulations. Note that the meshes were evenly sized. Therefore, the overall number of cells rises with a second power for 2D meshes from 6784 to 106,206 cells for 11 and 42 cells within $r_o$ distance, respectively, and with a third power for 3D meshes from 131,948 to 686,300 for 16 and 27 cells in $r_o$, respectively. The wall $y^+$ for the finest mesh in the range from 6 to 40 for *Re* = 2000. The 3D mesh with 35 cells uses a similar cell size as the 27 cells

version but contains 6 prismatic cells in the wall boundary layer to reach the wall $y^+ \sim 1$ along the exit orifice. Moreover, three different dimensions of the outer outflow area were also compared, but the effect on the results was negligible. Thus, the middle size area was used for all the cases.

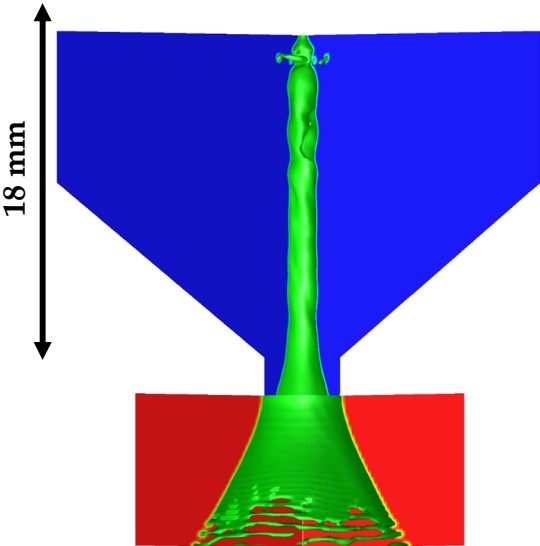

**Figure 7.** Typical results obtained from 3D periodic mesh with 27 cells in $r_o$, transient, Laminar simulation. $Re$ = 2000, instantaneous image. Blue and red represent liquid and air, respectively, and green is an interface.

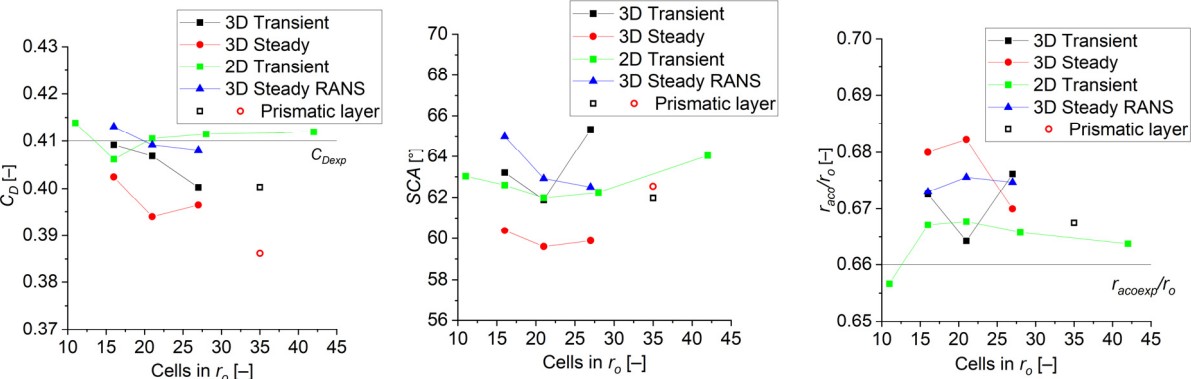

**Figure 8.** Mesh independence test: the effect of cell number in $r_o$. Laminar model, $Re$ = 2000.

The steady laminar simulation was found to be more prone to mesh sizing and reached a maximum difference in $C_D$ of 7%, while the RANS simulation showed lower dependency on cell size. The transient laminar simulation overperformed the steady one in proximity of results to the experimental data and showed a small effect of the mesh size. The sensitivity of the steady model can be explained by different VOF schemes; the steady simulations used a Compressive scheme, while the transient one used a more accurate geo-reconstruct scheme [30]. Moreover, the steady laminar simulations overestimated the air-core size, which leads to a decrease in $C_D$. The optimal mesh for both 2D and 3D has at least 20 cells in $r_o$, which is in line with other authors; however, a large variance was found in the literature. Ghate et al. [19] suggest a mesh with roughly 13 cells in $r_o$, Ghate et al. [19] used approximately 25 cells in $r_o$, Vashahi et al. [16] similarly used around 25 polyhedral cells in $r_o$ plus 7 cells of prismatic layer, while Galbiati et al. [12] found a mesh independence for a much finer mesh, with roughly 40–50 cells in $r_o$ and Nouri-Borujerdi et al. [14] used even finer mesh with 80 cells in $r_o$. However, the overall effect of the mesh size is rather small

and other factors, such as a turbulence model, time statistics or initial conditions, may have a greater effect.

The main advantage of the steady simulation is its time efficiency. It took between 5–9 h for mesh with 686,300 cells on a 12-cores machine (Intel® Xeon® Silver 4214 @ 2.2 GHz, Santa Clara County, CA, USA) to reach the quasi-steady state. The same case but transient took about 3–4 weeks to produce the same level of convergence. For practical usage, the steady simulations are much more suitable.

### 3.3. Atomizer Geometry Simplification

The atomizer geometry can be simplified in several steps; see Table 3. The most complex approach uses the full-scale atomizer model including the inlet ports. This model allows for the air-core to divert from the swirl chamber centreline. Since the atomizer used here can be divided into three identical parts, the periodic model can be easily created. This simplification ensures that the air-core axis is equal to the swirl chamber centreline, which may subdue some large-scale flow instabilities. This model can be further simplified, as the inlet port geometry can be substituted by an imprint of the inlet port on the wall of the swirl chamber to form the inlet boundary condition. However, the inlet velocity profile on the boundary of the swirl chamber is not trivial and it must be pre-calculated to yield correct results. This pre-calculation can be performed using a simple tetrahedron mesh with inlet port, a single-phase model and a steady solution. This velocity profile must be interpolated on the no-port inlet boundary. The main advantage of the no-port mesh is the absence of highly skewed cells near the swirl chamber; therefore, the mesh creation process is much faster and the mesh quality is better.

**Table 3.** Comparison of model geometry. *Re* = 2000, laminar model, transient.

| Mesh | $p_l$ | SCA | $d_{ac\text{-}exit}$ | $d_{a11mm}$ | $U_l$ | $V_l$ | $C_D$ |
|---|---|---|---|---|---|---|---|
| | [Pa] | [°] | [mm] | [mm] | [m/s] | [m/s] | [–] |
| Experimental data | 8800 | 78.0 | 2.99 | 1.67 | 3.1 | 2.2 | 0.41 |
| 2D Momentum-based | 6900 | 56.1 | 2.88 | 1.48 | 3.4 | 1.7 | 0.46 |
| 2D $v_p$ | 8800 | 62.2 | 3.06 | 1.64 | 3.5 | 2.2 | 0.41 |
| 3D PER No-port | 6700 | 56.7 | 2.83 | 1.47 | 3.5 | 1.9 | 0.47 |
| 3D PER No-port-profile | 8550 | 63.0 | 2.98 | 1.49 | 3.6 | 2.4 | 0.42 |
| 3D PER | 9200 | 65.3 | 3.07 | 1.85 | 2.9 | 2.6 | 0.40 |
| 3D Full | 9350 | 61.3 | 3.10 | 1.66 | 3.7 | 2.4 | 0.40 |

The last possible simplification is to create a 2D axisymmetric model with the swirl velocity component. This model requires setting the inlet radial and swirling velocity as two independent components to keep the momentums in both directions. However, the results show that conserving the angular momentum can be inaccurate due to distortion of the inlet velocity profile, which virtually increases the inlet angular momentum. This phenomenon is also well observed on the no-port version, which underestimated the air-core dimensions as well as the inlet pressure.

When the full 3D mesh is compared with the periodic one in terms of velocity profiles (Figure 9) and global parameters (Table 3), only minor differences can be found. In most practical calculations, the periodic mesh is more cost-effective. The no-port version did not prove prediction reliability even if the inlet velocity profile is pre-calculated and therefore is not recommended for further use.

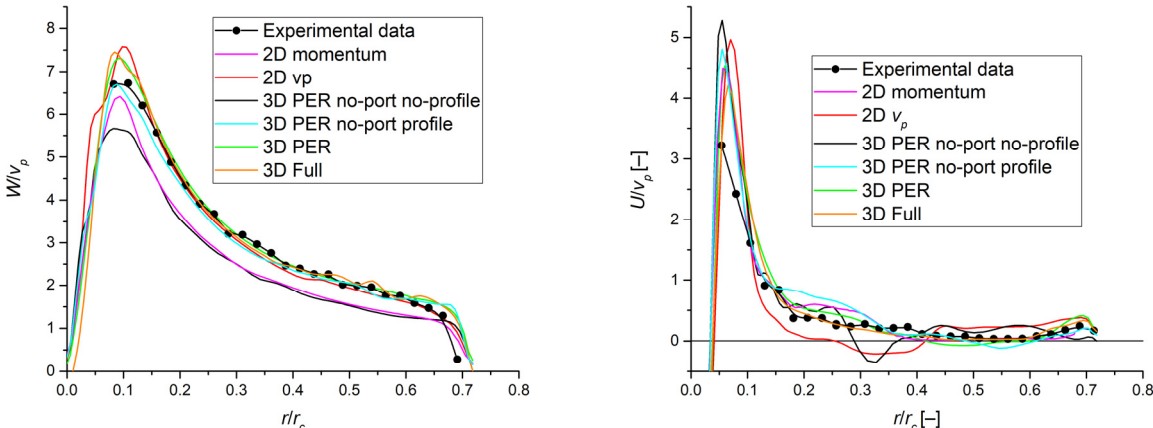

**Figure 9.** Mean swirl velocity profiles (**left**), axial velocity profiles (**right**). Z = 11 mm, *Re* = 2000.

### 3.4. Physical Models

Several physical models were compared for the 3D periodic mesh with the inlet port for all the regimes investigated. The results for both steady and transient simulations are listed in Table 4 for *Re* = 2000. The effect of *Re* on the results of the selected models is shown in Figure 10.

**Table 4.** Comparison of different physical models, 3D periodic mesh with port, *Re* = 2000.

| Mesh | | Physical Model | $p_l$ | SCA | $d_{ac\text{-}exit}$ | $d_{a11mm}$ | $U_l$ | $V_l$ | $C_D$ |
|------|---|----------------|-------|-----|--------------|-------------|-------|-------|-------|
| | | | [pa] | [°] | [mm] | [mm] | [m/s] | [m/s] | [–] |
| Structured | | Laminar | 9200 | 59.9 | 3.08 | 1.82 | 3.85 | 2.41 | 0.40 |
| Structured | Steady | $k$-$\varepsilon$—Realizable Scalable WF | 8950 | 53.6 | 3.10 | 1.72 | 3.70 | 2.31 | 0.41 |
| Structured | | RSM Scalable WF | 7400 | 59.9 | 2.51 | 0 | 2.69 | 1.66 | 0.45 |
| Prismatic layer | | $k$-$\omega$—SST | 10421 | 62.0 | 3.10 | 1.56 | 4.05 | 2.92 | 0.38 |
| Prismatic layer | | $k$-$\varepsilon$—Realizable Enhanced WT | 8870 | 58.0 | 2.97 | 1.45 | 3.02 | 2.19 | 0.41 |
| Prismatic layer | | $k$-$\varepsilon$—RNG Enhanced WT | 8800 | 57.2 | 2.95 | 1.40 | 2.99 | 2.16 | 0.41 |
| Prismatic layer | | RSM—Omega | 10210 | 58.3 | 3.08 | 1.49 | 3.69 | 2.62 | 0.38 |
| Prismatic layer | | RSM Enhanced WT | | | Diverged | | | | |
| Structured | | Laminar | 9300 | 65.3 | 3.07 | 1.85 | 2.92 | 2.64 | 0.40 |
| Structured | Transient | $k$-$\varepsilon$—Realizable Scalable WF | 8540 | 54.1 | 3.07 | 1.67 | 3.63 | 2.35 | 0.42 |
| Structured | | RSM Scalable WF | | | Diverged | | | | |
| Prismatic layer | | LES—WALE | 8960 | 59.9 | 3.03 | 1.55 | 3.69 | 2.64 | 0.41 |
| Prismatic layer | | $k$-$\omega$-SST | 8840 | 57.6 | 3.01 | 1.50 | 3.02 | 2.66 | 0.41 |
| Prismatic layer | | $k$-$\varepsilon$—Realizable Enhanced WT | 8050 | 52.0 | 2.87 | 1.22 | 3.11 | 2.25 | 0.43 |
| Prismatic layer | | RSM—Omega | 9100 | 54.0 | 3.02 | 1.56 | 3.07 | 2.62 | 0.41 |
| Prismatic layer | | RSM Enhanced WT | | | Diverged | | | | |

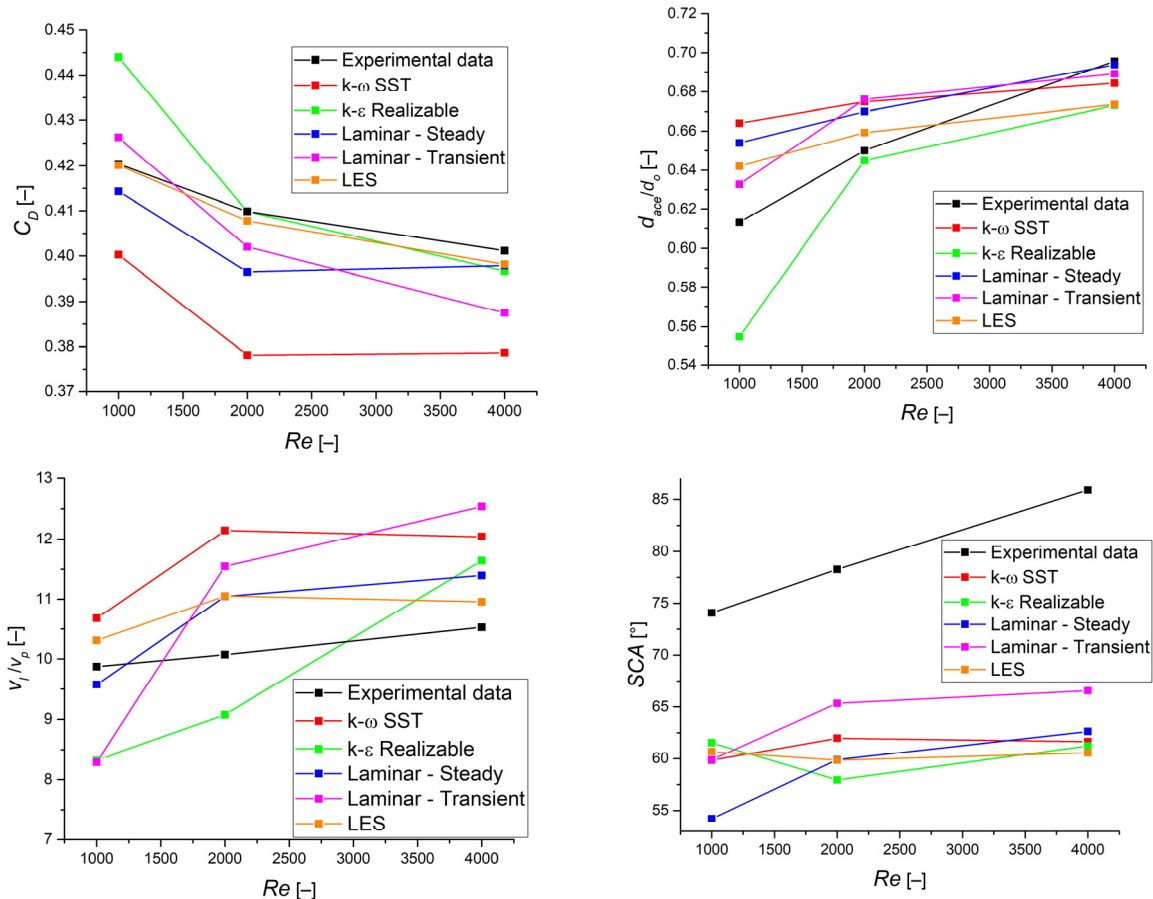

**Figure 10.** Effect of *Re* on $C_D$, *SCA*, liquid sheet velocity, $v_l$, and $d_{ace}$ from various numerical setups.

Similarly, as in the mesh independence test, the steady simulation was found prone to selecting the model. Both versions of steady *k-ε*, RNG and Realizable, performed almost identically and their results were close to the experimental data. The steady RSM reached the largest deviation from the experimental data since it was unable to predict the air-core within the swirl chamber. This model also failed in the mesh with a prismatic layer using Enhanced wall treatment. The same model, but omega-based RSM, properly captured the air-core but underestimated the $C_D$. Similar results were obtained by the steady *k-ω*–SST model. Both the *k-ω*–SST and omega-based RSM predicted $C_D$ accurately in the transient simulation, where all the turbulence models return virtually identical results including LES. Only the omega-based RSM converged in the transient simulations, other variants of RSM diverged.

An improper setup of the wall function results in failure of the air-core prediction. The wall $y^+$ must be carefully checked before selecting the proper wall function as the standard wall function failed in all investigated cases. The turbulent models were able to closely predict flow characteristics even for *Re* = 1000, which is in good agreement with [12], where the *k-ε* model returns reliable results for *Re* = 1600. Note here that the RSM model underperformed other turbulence models, similarly as in [12]. This result was not expected, since the RSM should be superior for flows with anisotropic turbulence, which is the case with swirl atomizers. It predicts the air-core only in the case of omega-based RSM, but its results were practically identical to the simpler *k-ω*.

The suitable models from Table 4 were compared in the range of *Re* and the results are presented in Figure 10. No RSM model was able to predict the air-core for *Re* = 1000, thus these models are not presented here.

The measured $C_D$ decreased slightly with *Re* as $C_D \propto Re^{-0.035}$, respectively with $p_l$ as $C_D \propto p_l^{-0.016}$. It is in good agreement with several authors [31,32] who found a similar

decrease in $C_D$ but some authors [33] claimed rather ascending $C_D$ with $p_l$. Nevertheless, all CFD models predicted the same downward trend, yet minor differences were observed. The steady $k$-$\omega$ model underpredicted the $C_D$, particularly for $Re = 1000$. The $k$-$\varepsilon$ also suffers from difficulties with this regime, where it severely overpredicts the $C_D$ and underpredicts the air-core dimension. The LES model returns the closest results of $C_D$, followed by the transient laminar model, which slightly diverts from the experimental data at $Re = 4000$. Nevertheless, the mean absolute percentage error (MAPE) of both laminar simulations, LES and $k$-$\varepsilon$ was below 2.5%. This indicates that the prediction of $C_D$ is robust regardless of the model used. Note here that the empirical correlation for $C_D$ proposed by Rizk and Lefebvre [34] gives a constant value of $C_D$ of 0.41, which is in perfect agreement with the experimental data for $Re = 2000$.

The $C_D$ value is usually related to the air-core diameter. Therefore, a decrease in $C_D$ should be accompanied by the growing air-core diameter, which is true for both experimental data and all the CFD models, see Figure 10, right. Only the $k$-$\varepsilon$ model diverts for $Re = 1000$. Other models tend to slightly overestimate the air-core size for low $Re$. The MAPE of $d_{ace}$ prediction of all the models is below 4.5%.

The relative velocity of discharged liquid sheet, $v_l/v_p$ (combined radial and axial velocity divided by the inlet velocity) is slightly increasing with $Re$. All CFD models captured the trend from the experiment well, but most of them overestimated the velocity values. The laminar model was closest to the experimental data for $Re = 1000$ and 2000, followed by the LES and $k$-$\varepsilon$. Slightly different results were obtained at the highest $Re$, where the LES outperformed the laminar model, but still overestimated the velocity by 5%. This results in relatively high MAPE values of 6% for LES a laminar simulation.

A huge disparity is found in the *SCA* results, where the value of the experimental *SCA* is more than 20% larger than the predicted one. The experimental *SCA* discussed here is measured directly after discharge at the apex angle of the cone, which covers the liquid sheet, while the *SCA* from the CFD is based on the maximum in the liquid fraction inside the discharge liquid sheet. Nevertheless, both approaches should return very similar values of *SCA*. The *SCA* grows significantly in the experiment with $Re$ as $SCA \propto Re^{0.12}$, respectively with $p_l$ as $SCA \propto p_l^{0.06}$, while it increases in transient laminar simulation as $SCA \propto Re^{0.06}$ or $SCA \propto p_l^{0.03}$ and it is constant for the LES simulation. The reason for these disparities is not clearly known, but may be partially related to the flow of the surrounding air, which may slightly affect the liquid sheet formation as described in [35] and was not captured in CFD due to a relatively small outflow area. Furthermore, manufacturing inaccuracies and different measuring methods may introduce some errors.

A widely used empirical correlation published by Rizk and Lefebvre [36] predicted a much higher influence of the inlet pressure as $SCA \propto p_l^{0.11}$. However, this correlation was derived for liquids with higher viscosity and well captured the trends using higher viscous liquid in our previous study [25] using almost identical geometry as used here. Similarly, the experimental values of *SCA* were higher than expected. The viscous liquids typically exhibit a larger influence of $p_l$ on *SCA* since Ballester et al. [33] found $SCA \propto p_l^{0.39}$ for heating oil. The values of *SCA* based on the Rizk predictions are 46, 54 and 63° for $Re = 1000$, 2000 and 4000, respectively. These values are well below the measured values but are much closer to the CFD prediction. Since the CFD correctly predicts other parameters of the internal flow, the differences in *SCA* might be linked to a different measurement technique, complex airflow near the liquid sheet, or wettability of the atomizer PMMA body.

### 3.5. Air-Core Dynamics

The mean parameters of the internal flow are important for the geometric design of the atomizer. However, instabilities are commonly presented and for some geometries, they may change the breakup nature or generate an unstable spray, as documented elsewhere [1,25,26]. The steady simulations, in principle, can only indicate the calculation stability or the convergence rate, which may or may not be linked with real flow instabilities. Therefore, this part focuses only on transient models. The simplest way to detect flow

instabilities is to measure the rate of change of the air-core diameter or its position. From the experimental data, the most prominent are the air-core contractions and extractions at a given position as seen in Figure 11. The same dominant frequencies were also observed for the air-core boundaries, but with slightly less prominence. Its frequencies, shown in Figure 12, are almost identical along the air-core length and peaked roughly at $f = 110$, 240 and 480 Hz for $Re = 1000$, 2000 and 4000, respectively. This leads to an almost constant value of the Strouhal number over the range of $Re$ values:

$$St = \frac{fD}{v_p} \tag{1}$$

where $D$ is a characteristic dimension, which can be the swirl chamber diameter and is constant here. The same behaviour was documented in [37], where the value of $St$ depended on the atomizer constant and was independent of the operating regime. Note here that no dominant frequency was observed in the measured LDA data.

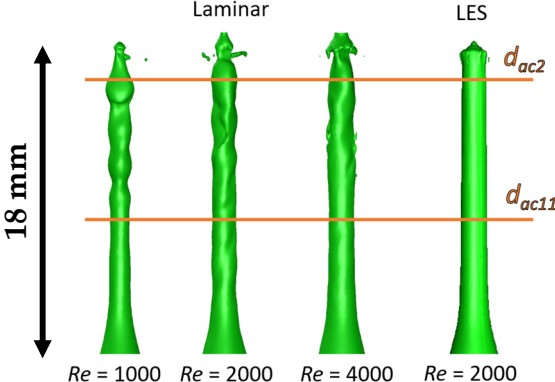

**Figure 11.** The air-core surface from laminar simulation and LES (right).

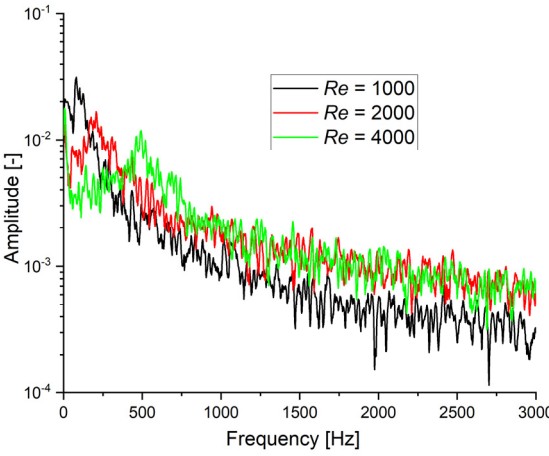

**Figure 12.** Frequency spectra of air-core diameter fluctuations at location $d_{ac11}$. Experimental data.

　　　The CFD time-resolved results did not match the experimental data well. The turbulence models, including the LES model, subdued the air-core surface waves and almost no fluctuations can be detected, see the smooth air-core structure in Figure 11. The laminar model exhibits the largest amplitudes of the air-core fluctuations as reported in Figure 13 where the frequencies for two locations, $d_{ac2}$ and $d_{ac11}$, are shown. Nevertheless, the air-core temporal behaviour is different from the experimental data, since no dominant frequencies and a rather smooth air-core surface is presented in the bottom part of the swirl chamber, see Figures 11 and 13 left. The top part of the air-core at the location $d_{ac2}$ exhibits periodical fluctuations with a dominant frequency of 470, 930, and 1900 Hz for $Re = 1000$, 2000, and 4000, respectively and the same frequencies are also observable for the swirl velocity

component near the air-core. The frequencies are roughly 4 times higher compared to the frequencies from the experiment, but it follows the same trend of constant *St*. The turbulence intensity of the swirl velocity, based on the velocity standard mean deviation, reached values from 0.5 to 2% for the laminar simulation. These values are roughly 4-times lower compared to the experimental data. It tends to increase towards the top of the swirl chamber and also with increasing *Re*, which is the opposite behaviour to the experiment.

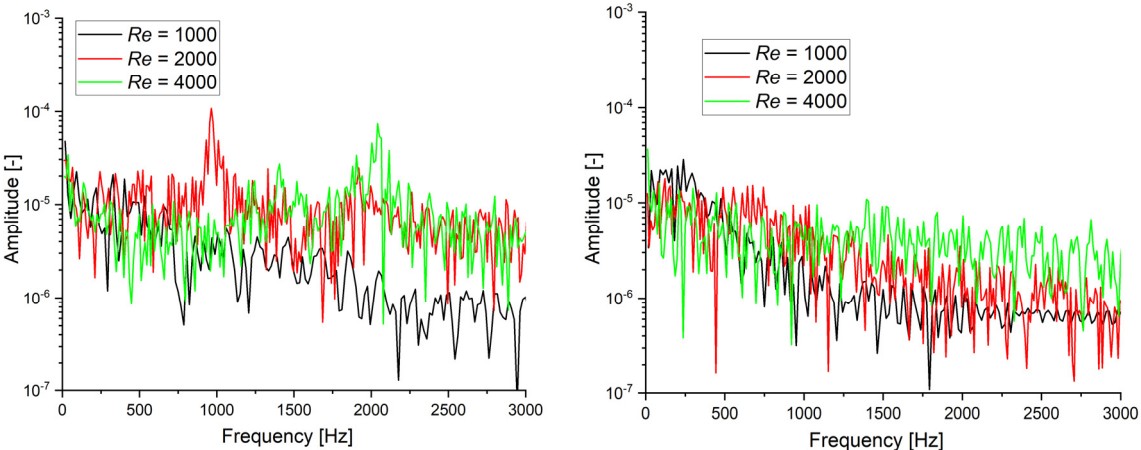

**Figure 13.** Frequency spectra of air-core diameter fluctuations at location (**left**): $d_{ac2}$, (**right**): $d_{ac11}$. Laminar simulation.

## 4. Conclusions

The 2D and 3D numerical simulations of pressure-swirl atomizer internal flow were compared in transient and steady states with experimental data for *Re* = 1000, 2000 and 4000. Various numerical setups, including six turbulence models and several geometrical simplifications, were investigated.

The air-core was found stable and developed in all the tested regimes. The discharge coefficient $C_D$ slightly decreased with *Re* as a result of lower relative viscous losses, which was confirmed by a slight increase in the measured relative swirl velocity.

The results from steady simulations are comparable with the time-averaged transient ones, but their convergence rate is at least about an order of magnitude higher. No steady simulation setup was found stable for the 2D model.

From the geometry simplification study, it is evident that the inlet port must be modelled in full to assure reliable results. The full 3D model can give only a little advantage compared to the periodical geometry.

The laminar solver was capable to predict the $C_D$, air-core dimensions, and velocity profiles with an error of less than 5% compared with the experimental results in both 2D and 3D simulations for the whole range of *Re*. The LES model performed similarly to the laminar solver for low *Re* and was slightly more accurate for *Re* = 4000. The two-equation models, *k-ε* and *k-ω*, were sensitive to proper solving of the near wall flow and were not accurate for low *Re*. However, all the models captured well the trends. The worst results were surprisingly obtained for RSM (Reynolds Stress model), which diverged for *Re* = 1000 and 2000 or predicted the undeveloped air-core. For an unknown reason, the only parameter predicted with a large error was the spray cone angle (*SCA*).

The air-core diameter fluctuations were observed and the dominant frequency was found to rise with *Re* to keep a constant value of the Strouhal number. However, the transient simulations did not match the frequencies and turbulence models suppressed the amplitude of the surface waves.

For the atomizer rapid CFD prototyping, the steady 3D simulation with a laminar solver is the most efficient approach. The transient simulations may be helpful in predicting unstable atomizers for some more extreme geometries, but experimental verification is still required.

**Author Contributions:** Conceptualization, M.M. and J.J.; methodology, M.M. and J.S.; software, J.S.; validation, J.S. and O.C.; investigation, M.M., J.S. and O.C.; data curation, M.M., J.S. and O.C.; writing—original draft preparation, M.M. and J.S.; writing—review and editing, M.M.; visualization, M.M.; supervision, J.J.; project administration, J.J.; funding acquisition, J.J. All authors have read and agreed to the published version of the manuscript.

**Funding:** This research was funded by the project "Computer Simulations for Effective Low-Emission Energy Engineering" funded as project No. CZ.02.1.01/0.0/0.0/16_026/0008392 by Operational Programme Research, Development and Education, Priority axis 1: Strengthening capacity for high-quality research.

**Institutional Review Board Statement:** Not applicable.

**Informed Consent Statement:** Not applicable.

**Data Availability Statement:** The experimental data that support the findings of this study are openly available here: https://1url.cz/WKLF9 (accessed on 19 June 2022).

**Conflicts of Interest:** The authors declare no conflict of interest.

## Nomenclature

| | |
|---|---|
| $A$ | area [m$^2$] |
| $b$ | width [m] |
| B | experimental constant [–] |
| $C_D$ | discharge coefficient [–] |
| $d$ | diameter [m] |
| $h$ | height [m] |
| $k$ | atomizer constant used in [4,5] [–] |
| $l_b$ | breakup length [m] |
| $\dot{m}$ | mass flow rate [kg/h] |
| $n$ | refractive index [–] |
| $r$ | radial distance [m] |
| $Re$ | Reynolds number [–] |
| $S_1$ | virtual distance of the measurement volume [m] |
| $S_2$ | real distance of the measurement volume [m] |
| $SCA$ | spray cone angle [°] |
| $SFR$ | spill-to-feed ratio [–] |
| $t$ | liquid sheet thickness [m] |
| $U$ | axial velocity [m/s] |
| $V$ | radial velocity [m/s] |
| $v$ | velocity [m/s] |
| $W$ | swirl velocity [m/s] |
| $We_g$ | gas Weber number [–] |
| $X$ | ratio of air-core to exit orifice area [–] |
| $Z$ | axial distance [m] |
| Greek characters | |
| $p$ | pressure drop at the nozzle [Pa] |
| $\mu$ | dynamic viscosity [kg/(m·s)] |
| $\rho$ | density [kg/m$^3$] |
| $\sigma$ | liquid/gas surface tension [kg/s$^2$] |
| Subscripts and Superscripts | |
| $c$ | swirl chamber |
| $cal$ | calculated |
| $e$ | exit orifice |
| $g$ | surrounding gas |
| $l$ | atomized liquid |
| $p$ | inlet port |
| $ac$ | air-core |
| $sc$ | spray cone |

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
