# Peer review of "Searching for a Numerical Model for Prediction of Pressure-Swirl Atomizer Internal Flow"

_applsci, doi:10.3390/app12136357_

Round 1

Reviewer 1 Report

This paper compares 2D (two-dimensional) axisymmetric, 3D (three-dimensional) periodic, and full 3D numerical models used to estimate the internal flow characteristics of a pressure-swirl atomizer. The paper has a novelty. The computation validation and the data verification using the experiment in this manuscript version are explained. The results and discussion among some CFD models are well described. Overall, the manuscript is worthy of publication. However, some crucial issues should be explained further. Therefore, I recommend the publication can be accepted after the following comments are considered in a revised version.

 1.      I would strongly advise the authors to present a schematic diagram of the experimental apparatus.

2.      Tested conditions covered a range of the Reynolds numbers within the inlet ports, Re = 1000, 2000, 4000. Please explain why? I think the CFD results depend on the relation between the CFD model and the setting of the Re number. Therefore, the limitations of pressure prediction from CFD simulation results should be explained.

3.      Figure 4 needs physical discussion.

4.      For validation, adding the mean absolute percentage error (MAPE) value for numerical and experiment benchmarking is better.

Author Response

First, we would like to thank the referees for their thorough reviews of our paper and their suggestions and recommendations.

We have carefully considered the reviewers’ suggestions and requirements and have attempted to satisfy them entirely, keeping in mind to improve the manuscript in a concise, space-saving manner.

Our responses are made (point by point) as raised in the reviewers’ comments and attached below each comment here.

Reviewer 1

This paper compares 2D (two-dimensional) axisymmetric, 3D (three-dimensional) periodic, and full 3D numerical models used to estimate the internal flow characteristics of a pressure-swirl atomizer. The paper has a novelty. The computation validation and the data verification using the experiment in this manuscript version are explained. The results and discussion among some CFD models are well described. Overall, the manuscript is worthy of publication. However, some crucial issues should be explained further. Therefore, I recommend the publication can be accepted after the following comments are considered in a revised version.

  1. I would strongly advise the authors to present a schematic diagram of the experimental apparatus.

We carefully considered this comment. However, the schematic diagram of the test bench is presented in the cited paper, in which the experiment is described in a greater detail. Unfortunately, there a copyright applies to that scheme. It would require long time to negotiate the usage of the scheme from the Journal office without a guarantee to acquire that.

  1. Tested conditions covered a range of the Reynolds numbers within the inlet ports, Re = 1000, 2000, 4000. Please explain why? I think the CFD results depend on the relation between the CFD model and the setting of the Re number. Therefore, the limitations of pressure prediction from CFD simulation results should be explained.

There were several considerations when selecting the proper operating regime. The Re = 4000 was chosen as the maximum flow rate of the liquid supply test bench and it was also close to the maximum pressure, which can atomizer handle without deformations. The Re = 1000 corresponds to the lowest pressure of the real-sized atomizer at which the atomizer generates developed spray. We also want to cover both the laminar regime (Re = 1000 and 2000) and transition to turbulence if any occurs (Re = 4000). The description of operating regime selection was improved in the revised manuscript.

There is definitely a relationship between the CFD model and Re and this is the aim of this paper. However, the CFD models are more sensitive to presence of turbulence, which, in this case, was similar for all cases, since transition to turbulence occurs probably at much higher values of Re.

  1. Figure 4 needs physical discussion.

The link and discussion of figure 4 was added into text.

  1. For validation, adding the mean absolute percentage error (MAPE) value for numerical and experiment benchmarking is better.

The mean absolute percentage error (MAPE) was add to the result description and discussion.

Reviewer 2 Report

The work presents an interesting study based on numerical simulations and experimental tests of the flow inside a pressure-swirl atomizer. The numerical simulations have been performed considering different conditions of the flow and different models for turbulence, comparing 2D and 3D cases. The paper is well written and it gives useful indication for the dimensioning of pressure-swirl atomizer.
There is only a concern about the simulations shown in the paper. What do the authors exactly mean with Laminar simulation? How it is possible to model Re number for the Laminar simulation greater than those of the LES simulation? In principle a LES simulation should simulate Re number which are greater than those observed in a laminar flow. Please clarify on that.
There is also minor suggestion about the presentation of the numerical method used, for which the author used the commercial software Ansys Fluent 19.2. Even if there are reference to other papers, the manuscript would be more readable if the fluid dynamic equation, together with the exact expressions of the RANSE turbulent closure used are presented in the section Numerical setup.

Author Response

First, we would like to thank the referees for their thorough reviews of our paper and their suggestions and recommendations.

We have carefully considered the reviewers’ suggestions and requirements and have attempted to satisfy them entirely, keeping in mind to improve the manuscript in a concise, space-saving manner.

Our responses are made (point by point) as raised in the reviewers’ comments and attached below each comment here.

Reviewer 2

The work presents an interesting study based on numerical simulations and experimental tests of the flow inside a pressure-swirl atomizer. The numerical simulations have been performed considering different conditions of the flow and different models for turbulence, comparing 2D and 3D cases. The paper is well written and it gives useful indication for the dimensioning of pressure-swirl atomizer.

There is only a concern about the simulations shown in the paper. What do the authors exactly mean with Laminar simulation? How it is possible to model Re number for the Laminar simulation greater than those of the LES simulation? In principle a LES simulation should simulate Re number which are greater than those observed in a laminar flow. Please clarify on that.

The problem is that the flow regime inside the atomizer is not known as it is difficult to find the right Re definition for the complex swirling flow. Different Re definition are used. And there is a problem of setting the threshold Re for laminar-turbulent transition. Some authors suggested laminar flow conditions up to Re = 40000, while others claimed to be turbulent even at Re = 2000. The laminar model solves directly Navier-Stokes equations without any turbulence or subgrid models. The simulation will not converge or return meaningless results for turbulent flow, due to limited mesh resolution and selected discretization scheme. While LES simulation, when applied to laminar flow and relatively fine mesh, should be close to the laminar model, but due to subgrid models, it will work properly on turbulent flows on coarse meshes. The model selection provided here illustrated that the flow should be treated as laminar up to Re = 4000.

There is also minor suggestion about the presentation of the numerical method used, for which the author used the commercial software Ansys Fluent 19.2. Even if there are reference to other papers, the manuscript would be more readable if the fluid dynamic equation, together with the exact expressions of the RANSE turbulent closure used are presented in the section Numerical setup.

We carefully considered this suggestion. Adding a detailed description of the governing equations and used models would add some readability for fluid flow experts, but the general audience is probably more focused on the results and the CFD application itself. Therefore, a lot of equations would worsen the clarity of the paper for a general audience and make it lengthy. We think that the governing equations are well and repeatedly described in cited literature which is available for free.

Reviewer 3 Report

This study uses a numerical prediction of discharge parameters of a pressure swirl atomizer. 2D and 3D simulations were performed with laminar as well as four turbulent models to be compared with Laser Doppler Anemometry. Authors described the experimental setup thoroughly but failed to address the grid independence study on the models. Once authors provide this explanation, I will be able to accept the paper. 

Author Response

First, we would like to thank the referees for their thorough reviews of our paper and their suggestions and recommendations.

We have carefully considered the reviewers’ suggestions and requirements and have attempted to satisfy them entirely, keeping in mind to improve the manuscript in a concise, space-saving manner.

Our responses are made (point by point) as raised in the reviewers’ comments and attached below each comment here.

Reviewer 3

This study uses a numerical prediction of discharge parameters of a pressure swirl atomizer. 2D and 3D simulations were performed with laminar as well as four turbulent models to be compared with Laser Doppler Anemometry.

Authors described the experimental setup thoroughly but failed to address the grid independence study on the models. Once authors provide this explanation, I will be able to accept the paper. 

The grid independence study was extended by the RANS realizable k-ε model with scalable wall function CFD model, see Figure 8 and the subsequent discussion. Since laminar simulation is the most widely used for simulations of the internal flow and also yield the best results, the emphasis is still placed on that one.